# Comparison of Recurrence Patterns between Adenocarcinoma and Squamous Cell Carcinoma after Stereotactic Body Radiotherapy for Early-Stage Lung Cancer

**DOI:** 10.3390/cancers15030887

**Published:** 2023-01-31

**Authors:** Nozomi Kita, Natsuo Tomita, Taiki Takaoka, Shuou Sudo, Yusuke Tsuzuki, Dai Okazaki, Masanari Niwa, Akira Torii, Seiya Takano, Akio Niimi, Akio Hiwatashi

**Affiliations:** 1Department of Radiology, Nagoya City University Graduate School of Medical Sciences, 1 Kawasumi, Mizuho-cho, Mizuho-ku, Nagoya 467-8601, Aichi, Japan; 2Department of Respiratory Medicine, Allergy and Clinical Immunology, Nagoya City University Graduate School of Medical Sciences, 1 Kawasumi, Mizuho-cho, Mizuho-ku, Nagoya 467-8601, Aichi, Japan

**Keywords:** stereotactic body radiotherapy, non-small-cell lung cancer, squamous cell carcinoma, adenocarcinoma, local neoplasm recurrence

## Abstract

**Simple Summary:**

Stereotactic body radiotherapy (SBRT) is an effective modality for early-stage lung cancer. However, limited information is currently available on histological differences in recurrence patterns after SBRT for early-stage lung cancer. Therefore, the present study compared recurrence patterns between adenocarcinoma (ADC) and squamous cell carcinoma (SCC) after SBRT. Among the 204 patients included in the analysis, 138 and 66 were in the ADC and SCC groups, respectively. The local recurrence (LR) rate was significantly higher in the SCC group than in the ADC group, while lymph node metastasis and distant metastasis rates were not associated with the histological type. Tumor diameter and histological type correlated with LR in multivariate analyses. The present results suggest that the risk of LR after SBRT is higher for SCC than for ADC.

**Abstract:**

We compared recurrence patterns between adenocarcinoma (ADC) and squamous cell carcinoma (SCC) after stereotactic body radiotherapy (SBRT) for early-stage lung cancer. Patients with ADC and SCC histology, who were treated with SBRT for clinical stage IA1-IIA lung cancer at our institution, were included in the analysis. The rates of disease-free survival (DFS), overall survival (OS), local recurrence (LR), lymph node metastasis (LNM), and distant metastasis (DM) were calculated using the Kaplan–Meier method or the cumulative incidence function. Among the 204 patients analyzed, 138 and 66 were in the ADC and SCC groups, respectively. The median follow-up period was 60 months. The five-year DFS and OS rates were 57% vs. 41% and 69% vs. 48% in the ADC and SCC groups, respectively (*p* = 0.015 and 0.019, respectively). In the multivariate analysis, the histological type was not associated with DFS or OS. Five-year LR, LNM, and DM rates were 10% vs. 24%, 12% vs. 20%, and 25% vs. 27% in the ADC and SCC groups, respectively (*p* = 0.0067, 0.074, and 0.67, respectively). The multivariate analysis identified the histological type of SCC as an independent factor for LR (hazard ratio, 2.41; 95% confidence interval, 1.21–4.77; *p* = 0.012). The present results suggest that the risk of LR after SBRT is higher for SCC than for ADC.

## 1. Introduction

Lung cancer is the most common malignant tumor and the leading cause of cancer-related mortality worldwide. Non-small-cell lung cancer (NSCLC) accounts for an estimated 85% of all lung cancers, and adenocarcinoma (ADC) is the most common histological subtype of NSCLC, followed by squamous cell carcinoma (SCC). Computed tomography (CT) scan lung cancer screening is now widespread and has increased the detection rate of early-stage NSCLC to approximately one in four patients [1].

The standard treatment for medically operable early-stage NSCLC is currently surgical resection, and stereotactic body radiotherapy (SBRT) is typically performed for inoperable cases. However, emerging evidence suggests that SBRT is also a treatment option for operable stage I NSCLC [2,3,4]. The number of patients of an advanced age with impaired organ function and/or serious medical complications who are ineligible for thoracic surgery is increasing [5]. Therefore, appropriate SBRT management is becoming increasingly important for early-stage NSCLC. Retrospective cohort studies investigated recurrence patterns after SBRT [6,7,8,9], and the findings revealed differences compared to those after surgery [8,9]. There has been a limited number of studies investigating recurrence patterns after SBRT. We paid special attention to histological differences in NSCLC, which could be a risk factor affecting recurrence after SBRT. Therefore, the present study compared recurrence patterns between ADC and SCC after SBRT to assess its efficacy as a therapeutic strategy.

## 2. Materials and Methods

### 2.1. Patient Selection

The present study reviewed our single-institutional database of patients with early-stage NSCLC treated by SBRT between February 2004 and September 2018. Eligibility criteria were defined as follows: (1) histologically confirmed ADC or SCC; (2) clinical Tis-T2bN0M0 according to the 8th TNM classification [10]; (3) written informed consent provided. A total of 245 patients with early-stage NSCLC were treated with SBRT between February 2004 and September 2018 in Nagoya City University Hospital. Forty-one patients not pathologically diagnosed with ADC or SCC were excluded. Twelve patients with prior thoracic radiation therapy and 28 with a history of lung cancer surgery were also included in the study population. Among the 204 patients analyzed, 138 and 66 were in the ADC and SCC groups, respectively. Figure 1 shows the algorithm used to select patients for the study cohort. The present study was approved by the Institutional Review Board of Nagoya City University Graduate School of Medical Sciences (approval number: 60-22-0024). Since the requirement for written informed consent was waived due to the retrospective nature of this study, its content was disclosed in the form of an opt-out available on the website. This study followed the ethical standards laid down in the 1964 Declaration of Helsinki and its later amendments.

### 2.2. Pretreatment Evaluation

Clinical staging was based on a CT scan of the chest and upper abdomen, magnetic resonance imaging (MRI) or a CT scan of the brain, and 18F-fluoro-deoxyglucose positron emission tomography (FDG-PET). Since FDG-PET was not absolutely necessary, bone scintigraphy was also used. One hundred and forty-six patients were staged by FDG-PET, 38 by CT scan and bone scintigraphy and 20 by CT scan. Although staging by CT scan alone is not a standard approach, this staging method was also performed because some patients had trouble paying for FDG-PET or could not book an FDG-PET because the examinations were fully booked. The indication for surgery was determined by a tumor board consisting of respiratory physicians, respiratory surgeons, diagnostic radiologists, and radiation oncologists.

### 2.3. SBRT

Planning procedures were described in detail in previous studies [11,12]. The BodyFIX system (Medical Intelligence, Schwabmuenchen, Germany) was used for immobilization. Three phases of CT scan images with a 2.5 mm slice thickness were acquired for SBRT planning: normal breathing, the expiratory phase, and inspiratory phase. Three-dimensional treatment planning systems (Eclipse: Varian Medical Systems, Palo Alto, USA, or RayStation: RaySearch Medical Laboratories AB, Stockholm, Sweden) were used for SBRT planning. The gross tumor volume (GTV) was defined as the visible tumor based on CT scan and/or FDG-PET. The clinical target volume (CTV) was equal to the GTV. In addition, fluoroscopy was used to evaluate the respiratory motion of the tumor. The internal target volume (ITV) was created to cover the CTV in all respiratory phases. Additional anisotropic margins of 5 mm in the laterally and anteroposteriorly directions and 5–10 mm in the craniocaudal directions were added to the ITV to create the planning target volume (PTV). Five patients with large respiratory motion were irradiated during breath hold using metallic markers.

SBRT was performed with CLINAC 23EX between February 2004 and July 2014, with CLINAC 21EX between August 2014 and May 2015, and with TrueBeam from Jun 2015 (all Varian Medical Systems, Palo Alto, CA, USA). Regarding the patient alignments, the megavoltage portal imaging with CLINAC 23EX or 21EX was performed during all treatments. In addition, CT scans were acquired to evaluate the volume changes in targets in the first and third treatments. In the treatment with TrueBeam, cone-beam CT scan was used at each treatment for patient alignments.

Planned doses were prescribed to the isocenter of the PTV with a photon beam of 6 MV, and the prescribed dose was based on the diameter of the tumor. SBRT was administered twice weekly in 4 fractions and was performed at intervals of 3 days or longer based on radiobiological considerations [13]. As a rule, each treatment was spaced at least 72 h apart; however, due to patient schedule availability and machine availability, the actual treatment duration had a median of 12 days. The coverage of 95% of the PTV by at least 90% of the isocenter dose was recommended. Planned doses of 44, 48, and 52 Gy were prescribed for peripheral tumors with a maximum diameter of less than 1.5 cm, 1.5–3 cm, and larger than 3 cm, respectively, until November 2008. After December 2008, the protocol was changed for dose escalation and planned doses of 48, 50, and 52 Gy were administered according to each respective tumor diameter. Doses of 60 or 64 Gy in 8 fractions were used in cases of proximity to the pulmonary hilum or vital organs on an individual basis [14]. The dose calculation algorithm was pencil beam convolution with Batho power law between February 2004 and November 2008, the analytical anisotropic algorithm between December 2008 and May 2015, and collapsed cone convolution from June 2015.

### 2.4. Follow-Up and Collection and Evaluation of Data

After SBRT, CT scans were performed at 2- or 3-month intervals until 6 months. Thereafter, a chest and upper abdominal CT scan was performed at least every 6 months. FDG-PET and MRI or CT scans of the brain were performed whenever necessary. Local recurrence (LR) was diagnosed using serial CT scans combined with FDG-PET and/or biopsy. Dissemination to the pleura was considered to be distant metastasis (DM).

### 2.5. Statistical Analysis

Disease-free survival (DFS), cancer-specific survival (CSS), and overall survival (OS) were calculated by the Kaplan–Meier method. DFS was defined as the time from the start date of SBRT to any recurrence or death as events, and was censored at the last date without events. OS was defined as the time from the start date of SBRT to the last follow-up or death from any cause. LR was defined as the time from the start date of SBRT to recurrence within the primary tumor. The rates of LR, lymph node metastasis (LNM), and DM were calculated with a cumulative incidence function, and death was considered a competing risk.

To identify potential influencing factors, univariate and multivariate analyses were performed using the Cox proportional hazards model for DFS, CSS, and OS and with the Fine and Gray proportional hazards model for LR, LNM, and DM. All statistical analyses were carried out using EZR (Saitama Medical Center, Jichi Medical University, Saitama, Japan), which is a graphical user interface based on R (The R Foundation for Statistical Computing, Vienna, Austria) [15]. A *p*-value of <0.05 was considered significant.

## 3. Results

### 3.1. Patients

Patient characteristics are shown in Table 1. All the patients completed planned SBRT. The median follow-up time was 48 months (range 0–198) for all patients and 60 months (range 0–198) for living patients. The percentages of males and smokers (current or ex-smokers) were higher in the SCC group than in the ADC group (both *p* < 0.001). Tumor sizes were larger in the SCC group than in the ADC group (*p* < 0.001). Although the total dose was higher in the SCC group than in the ADC group (*p* = 0.041), the biological effective dose (BED) calculated with an α/β value of 10 was similar among the two groups. The median follow-up times of the ADC and SCC groups were 56 (range, 0–198) and 38 months (range, 1–188), respectively. One hundred and twenty-eight patients (63%) were inoperable. The reasons for this were pulmonary function impairment in 50 cases, cardiac disease in 20 cases, advanced age in 18 cases, previous pulmonary surgery in 16 cases, cerebral infarction in 8 cases, and other in 17 cases.

### 3.2. Outcomes

Of the 204 eligible patients, 79 (39%) died. Of these, 38 patients (19%) died of SBRT-treated lung cancer, while 41 (20%) died of other diseases. Deaths from other diseases were as follows: other cancers, 11; pneumonia, 8; heart disease, 7; others, 8; unknown, 7. Five-year DFS, CSS, and OS rates for all patients were 52% (95% confidence interval [CI], 44–59), 79% (95% CI, 71–85), and 63% (95% CI, 55–70), respectively. The median survival time was 8.4 years (95% CI, 6.3-NA). Figure 2A shows the survival curves of DFS, CSS, and OS for all patients. Five-year DFS, CSS, and OS rates in the ADC vs. SCC groups were 57% (95% CI, 48–65) vs. 41% (95% CI, 28–53) (*p*  =  0.015), 83% (95% CI, 75–89) vs. 67% (95% CI, 50–79) (*p * =  0.15), and 69% (95% CI, 60–77) vs. 48% (95% CI, 33–61) (*p*  =  0.019), respectively. Figure 2B–D show a comparison of the survival curves of DFS, CSS, and OS between the ADC and SCC groups.

Recurrence was detected in 77 (38%) of 204 eligible patients. A total of 6 cases were diagnosed with recurrence by pathology, 41 by FDG-PET, 1 by bone scintigraphy, and 29 by CT scan. In the ADC group, 3 cases were diagnosed with recurrence by pathology, 23 by FDG-PET, 1 by bone scintigraphy, and 21 by CT scan. In the SCC group, 3 cases were diagnosed with recurrence by pathology, 18 by FDG-PET, and 8 by CT scan. Cases with poor general conditions were diagnosed with recurrence by CT scan. A common type of failure was DM, which occurred in 41 cases (20%), followed by LR in 28 (14%), and LNM in 22 (11%). The DM site was the lungs in 19 cases (9%), pleural dissemination in 9 (4%), the brain in 4 (2%), the adrenal gland in 4 (2%), bone in 2 (1%), lung and pleural dissemination simultaneously in 1 (0.5%), lung, pleural dissemination and rib bone simultaneously in 1 (0.5%), and the bile duct in 1 (0.5%). The most common combination of recurrence was DM alone in 35 cases, followed by LR alone in 19. Among 77 cases with recurrence, 21 (27%) developed within one year after SBRT (6 cases of LR, 5 of LNM, 6 of DM, 1 of simultaneous LR and LNM, 1 of simultaneous LR and DM, and 2 of simultaneous LNM and DM). Eleven cases were diagnosed with recurrence by FDG-PET, one by bone scintigraphy, five by CT scan, and four by comprehensive evaluation. Among 77 patients with recurrence, 10 (13%) developed more than 5 years after SBRT. Between 5 and 10 years after SBRT, two cases of LR, one of LNM, three of DM, and one of simultaneous LR, LNM, and DM were observed. Two cases of DM and one of simultaneous LR and LNM were noted 10 years after SBRT.

The five-year LR and total recurrence rates for all patients were 15% (95% CI, 9.9–20) and 36% (95% CI, 29–43), respectively. Five-year LR, LNM, and DM rates in the ADC vs. SCC groups were 10% (95% CI, 5.8–17) vs. 24% (95% CI, 14–36) (*p* = 0.0067), 12% (95% CI, 6.9–18) vs. 20% (95% CI, 11–31) (*p* = 0.074), and 25% (95% CI, 18–33) vs. 27% (95% CI, 17–39) (*p* = 0.67), respectively. Figure 3A, B shows the survival curves of LR, LNM, and DM for the ADC and SCC groups. In the ADC group, DM was the most common, with a significant difference from LR and LNM (*p* = 0.0011). In the SCC group, no significant differences were observed in the recurrence type (*p* = 0.71). Figure 3C shows a comparison of survival curves of LR between the ADC and SCC groups (*p* = 0.0067).

### 3.3. Univariate and Multivariate Analyses of Outcomes

The results of univariate and multivariate analyses of DFS and OS are shown in Table 2. PS and tumor diameter correlated with DFS in both analyses. Age, sex, PS, and tumor diameter correlated with OS in both analyses.

The results of univariate and multivariate analyses of LR are shown in Table 3. Tumor diameter and histological type correlated with LR in both analyses. Regarding LNM and DM, no factor was associated with outcomes in both analyses (Table A1).

The results of univariate and multivariate analyses of LR in the tumor diameter ≤ 2.5 cm and >2.5 cm groups are shown in Table 4. In the tumor diameter ≤ 2.5 cm group, PS correlated with LR in both analyses. In the tumor diameter >2.5 cm group, histological type correlated with LR in the multivariate analysis. The risk of LR was higher in the SCC group than in the ADC group (hazard ratio [HR], 2.61; 95% CI, 1.07–6.41; *p* = 0.036).

## 4. Discussion

The present study compared recurrence patterns between ADC and SCC after SBRT for early-stage lung cancer and revealed a marked difference. The LR rate was higher in the SCC group than in the ADC group. The multivariate analysis revealed that the histological type of SCC was an independent factor for LR (HR, 2.41; 95% CI, 1.21–4.77; *p* = 0.012). Since tumor diameter correlated with LR, we performed a subgroup analysis using a tumor size of 2.5 cm as the cut-off value. This subgroup analysis also showed that SCC was an independent factor for LR in the >2.5 cm group (HR, 2.61; 95% CI, 1.07–6.41; *p* = 0.036). Similar to our results, several studies showed that SCC was a risk factor for LR after SBRT for early-stage lung cancer [7,16]. SCC is generally more radiosensitive than other histological types among various cancers [17,18,19]. However, the present study suggested that the radiosensitivity of SCC was lower than that of ADC in patients with early-stage lung cancer treated with SBRT. Ceppi et al. reported that the expression of the thymidylate synthase (TS) gene differed between SCC of the lung and ADC of the lung [20]. Due to the higher expression of TS in SCC, SCC may have a greater ability to repair DNA damage than ADC. D’Angelillo et al. showed that programmed cell death protein ligand 1 was highly expressed in SCC and that SCC was more immunosuppressive [21]. ADC and SCC of the lung have different percentages of tumors exhibiting p53 mutations. Previous studies showed that the percentage of p53 mutations was higher in SCC than in ADC [22,23]. Tumors with p53 mutations are more resistant to radiotherapy, which may be another reason for the higher rate of LR in the SCC group. Furthermore, radiation may be less effective against SCC of the lung; however, the underlying mechanisms remain unclear.

The frequency of LR was higher in the SCC group than in the ADC group. This result suggests that a therapeutic strategy for local control is more important for cancer control in the SCC group than in the ADC group. The use of different prescription doses according to tumor histology is not specified in the guidelines and is a new finding of this study [24]. The median planned dose and BED of this study was 50 Gy in 4 fractions and 113 Gy. In this study, the planned doses were prescribed to the isocenter of the PTV. It should be noted that the current SBRT guidelines generally prescribe PTV D95-D99 [24]. One promising therapeutic strategy for local control is a dose escalation to the PTV. Previous studies suggested that local control improved at higher doses. Parzen et al. reported that treatment with BED10 >150 Gy was associated with higher OS in patients with SCC of the lung [25]. Abel et al. suggested that BED10 >122 Gy was an independent prognostic factor for better OS in patients with SCC of the lung, and they argued that a new protocol needs to be investigated for SBRT for early-stage lung cancer that prescribes high BED values to the PTV while respecting the dose constraints of normal tissues [26]. Recurrence patterns were compared between ADC and SCC after chemoradiotherapy in patients with stage III lung cancer [27,28], and based on the findings, dose escalations were recommended to improve local control. In SBRT, the goal of which is local control, dose escalations may be recommended for patients with larger SCC tumors. However, the result of phase III RCT (RTOG 0617) showed increased adverse effects due to dose escalation. Dose escalation for lung cancer should be considered more carefully [29].

In future treatment strategies, adjuvant chemotherapy could be recommended to improve the outcomes of patients with ADC of the lung after SBRT because DM is common in these patients. Molecularly targeted drugs are currently not recommended after SBRT for early-stage lung cancer. However, the use of EGFR inhibitors after SBRT has potential, particularly for ADC with EGFR mutations. The ADAURA trial is currently examining the effects of EGFR inhibitors after surgery for localized lung cancer, including early-stage lung cancer, and has reported positive outcomes [30]. The Pacific trial showed that durvalumab after chemoradiotherapy for stage III lung cancer reduced the incidence of DM [31]. PDL-1 inhibitors may be recommended for patients with ADC prone to DM, even after SBRT. SBRT is also more likely to induce immunological changes than conventional irradiation and may be more effective [32,33,34,35]. An important issue involves the appropriate schedules of SBRT to enhance treatment effects and reduce toxicities. Our SBRT schedule was elaborated based on the findings of murine tumors models. We consider reoxygenation to be most important in enhancing the RT effect, and reoxygenation continues for more than 3 days. The strategy of using tumor reoxygenation to enhance RT effect was adopted in the SBRT schedule for NSCLC [36,37]. Therefore, the SBRT period was longer than in other institutions. The combined effect of RT with immunotherapy in patients with cancers is an increasing concern, and phase III clinical trials for various cancers, including SBRT for early-stage lung cancers, are ongoing [38]. The results of a Pacific-4 trial (RTOG 3515) evaluating the efficacy of durvalumab combined with SBRT for early-stage NSCLC will have an important impact. A study in murine tumors suggested that hypofractionated radiation using a few doses in the range of 8–12 Gy per fraction activated the immune pathway induced by DNA damage to irradiated cancer cells [39]. The SBRT period may affect DNA damage and repair, especially when combined with immunotherapy.

Our current study represents part of the last 15 years of SBRT outcomes at our institution. Our SBRT methods were almost in line with the ASTRO Evidence-Based Guideline for SBRT indications, pretreatment evaluation, and how to deal with central tumors, etc. [40]. Although the ASTRO guideline recommends that SBRT is appropriate for medically inoperable patients with T1-2N0 NSCLC, some operable patients were treated with SBRT in our institution. A multidisciplinary conference confirmed the appropriateness of the SBRT policy for operable patients who wished to undergo SBRT. In cases of proximity to the pulmonary hilum or vital organs, the number of SBRT fractions was increased at our institution. This treatment strategy in high-risk clinical scenarios was also consistent with the ASTRO guidelines.

There are some limitations that need to be addressed. The protocol was changed in December 2008 to increase the dose, and the analysis included different protocols. Furthermore, this was a retrospective study in a single institution. There was a potential bias in patient and treatment characteristics between the groups; however, most risk factors for recurrence were similar between the two groups. Moreover, a longer follow-up period is needed to observe recurrence patterns.

## 5. Conclusions

In conclusion, recurrence patterns were examined and compared between ADC and SCC in early-stage NSCLC after SBRT. Although DFS and OS were better in the ADC group than in the SCC group, the histological type was not associated with DFS or OS in the multivariate analysis. The frequency of LR was higher in the SCC group than in the ADC group, while the histological type was not associated with the incidence of LNM or DM. We found that the risk of LR after SBRT was higher for SCC than for ADC. It may be necessary to change the treatment approach depending on the histological type of early-stage NSCLC.

## Figures and Tables

**Figure 1 cancers-15-00887-f001:**
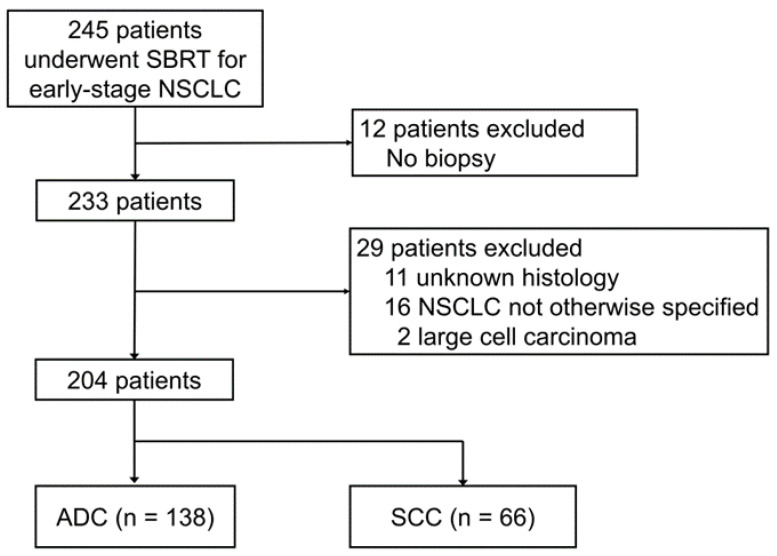
Algorithm for the study cohort.

**Figure 2 cancers-15-00887-f002:**
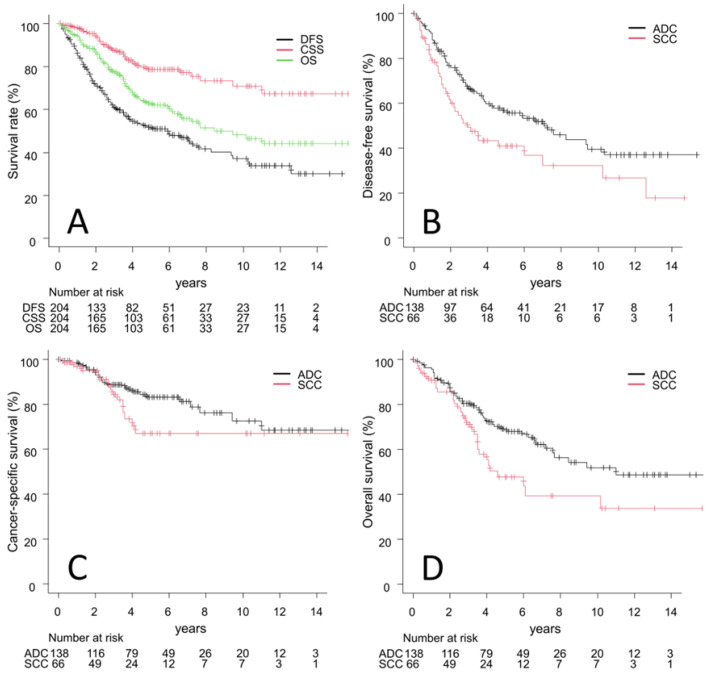
(**A**) Survival curves of disease-free survival (DFS), cancer-specific survival (CSS), and overall survival (OS) in all patients. (**B**–**D**) Comparisons of DFS (**B**), CSS (**C**), and OS (**D**) between adenocarcinoma (ADC) and squamous cell carcinoma (SCC).

**Figure 3 cancers-15-00887-f003:**
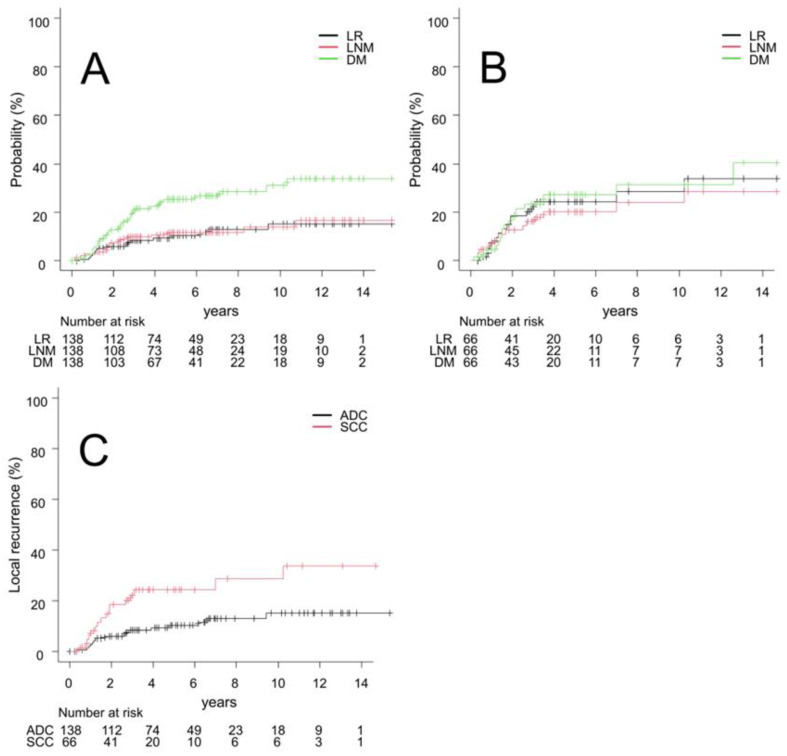
(**A**) Survival curves of local recurrence (LR), lymph node metastasis (LNM), and distant metastasis (DM) in adenocarcinoma (ADC). (**B**) Survival curves of LR, LNM, and DM in squamous cell carcinoma (SCC). (**C**) Comparison of LR curves between ADC and SCC.

**Table 1 cancers-15-00887-t001:** Patient and treatment characteristics.

Characteristics	All (*n* = 204)	ADC Group (*n* = 138)	SCC Group (*n* = 66)	*p* Value
**Age (years)**	77 (29–89)	77 (29–89)	78 (58–89)	0.25
**Male/female**	142 (70%)/62 (30%)	85 (62%)/53 (38%)	57 (86%)/9 (14%)	<0.001
**PS 0/1/2/3**	98 (48%)/85 (42%)/17 (8%)/4 (2%)	69 (50%)/59 (43%)/8 (6%)/2 (1%)	29 (44%)/26 (39%)/9 (14%)/2 (3%)	0.23
**Current smoker/ex/non/missing**	60 (29%)/88 (43%)/50 (25%)/6 (3%)	30 (22%)/56 (41%)/48 (35%)/4 (3%)	30 (45%)/32 (48%)/2 (3%)/2 (3%)	<0.001
**FEV_1_ (L)**	1.7 (0.6–3.3)	1.7 (0.7–3.3)	1.6 (0.6–3.0)	0.28
**Tumor diameter (cm)**	2.4 (0–5.0)	2.2 (0–4.7)	2.6 (0.9–5.0)	<0.001
**Tis/T1mi/T1a/T1b/T1c/T2a/T2b**	4 (2%)/5 (2%)/13 (6%)/47 (23%)/82 (40%)/42 (21%)/11 (5%)	4 (3%)/5 (4%)/12 (9%)/35 (25%)/51 (37%)/26 (19%)/5 (4%)	0 (0%)/0 (0%)/1 (2%)/12 (18%)/31 (47%)/16 (24%)/6 (9%)	0.041
**Tumor location**				
**Upper lobe/middle or lower lobe**	116 (57%)/88 (43%)	74 (54%)/64 (46%)	42 (64%)/24 (36%)	0.23
**Central/peripheral**	29 (14%)/175 (86%)	16 (12%)/122 (88%)	13 (20%)/53 (80%)	0.18
**Total dose (Gy)**	50 (44–64)	50 (44–64)	50 (48–60)	0.041
**Fractions**	4 (4–8)	4 (4–8)	4 (4–8)	0.16
**Biological effective dose (α/β = 10)**	113 (92–110)	113 (92–120)	113 (105–120)	0.23
**Operable/inoperable/missing**	74 (36%)/128 (63%)/2 (1%)	53 (38%) /84 (61%) /1 (1%)	21 (32%) /44 (67%) /1 (2%)	0.47

PS = performance status, FEV_1_ = forced expiratory volume in one second, ADC = adenocarcinoma, SCC = squamous cell carcinoma. Data are shown as *n* (%) or medians (range).

**Table 2 cancers-15-00887-t002:** Univariate and multivariable analyses of disease-free survival and overall survival.

	Disease-Free Survival				Overall Survival			
	Univariate		Multivariate		Univariate		Multivariate	
	HR (95% CI)	*p*-Value	HR (95% CI)	*p*-Value	HR (95% CI)	*p*-Value	HR (95% CI)	*p*-Value
Age (per year)	1.03 (1.0–1.06)	0.059	1.02 (0.99–1.04)	0.24	1.05 (1.02–1.09)	0.005	1.03 (1.0–1.07)	0.037
Sex (male vs. female)	1.78 (1.13–2.79)	0.012	1.55 (0.96–2.49)	0.075	2.44 (1.39–4.28)	0.002	2.04 (1.13–3.68)	0.018
PS (2, 3 vs. 0, 1)	2.14 (1.19–3.85)	0.011	1.91 (1.05–3.47)	0.034	2.48 (1.27–4.86)	0.008	2.15 (1.08–4.29)	0.030
FEV_1_ (L) (≤1.5 vs. >1.5)	0.87 (0.58–1.29)	0.47	0.86 (0.57–1.29)	0.47	0.74 (0.46–1.19)	0.21	0.73 (0.45–1.18)	0.20
Tumor diameter (per 0.1 cm)	1.04 (1.02–1.05)	<0.001	1.03 (1.01–1.054)	0.004	1.05 (1.03–1.07)	<0.001	1.05 (1.02–1.08)	<0.001
Histological type (SCC vs. ADC)	1.63 (1.09–2.42)	0.017	1.23 (0.81–1.88)	0.34	1.72 (1.09–2.71)	0.021	1.11 (0.68–1.81)	0.67
Biological effective dose (≤110 vs. >110)	1.20 (0.81–1.77)	0.37	0.91 (0.60–1.39)	0.67	1.10 (0.71–1.72)	0.67	0.77 (0.47–1.25)	0.28

PS = performance status, SCC = squamous cell carcinoma, ADC = adenocarcinoma.

**Table 3 cancers-15-00887-t003:** Univariate and multivariate analyses of local recurrence.

	Univariate		Multivariate	
	HR (95% CI)	*p*-Value	HR (95% CI)	*p*-Value
Age (per year)	0.98 (0.94–1.03)	0.39	0.97 (0.93–1.02)	0.23
Sex (male vs. female)	1.04 (0.51–2.13)	0.92	0.87 (0.43–1.75)	0.70
PS (2, 3 vs. 0, 1)	0.31 (0.043–2.30)	025	0.25 (0.033–1.81)	0.17
FEV_1_ (L) (≤1.5 vs. >1.5)	1.62 (0.82–3.21)	0.17	1.44 (0.73–2.84)	0.29
Tumor diameter (per 0.1 cm)	1.05 (1.02–1.08)	0.002	1.05 (1.01–1.08)	0.009
Histological type (SCC vs. ADC)	2.52 (1.27–4.99)	0.008	2.41 (1.21–4.77)	0.012
Biological effective dose (≤110 vs. >110)	1.34 (0.65–2.74)	0.43	0.95 (0.42–2.16)	0.90

PS = performance status, SCC = squamous cell carcinoma, ADC = adenocarcinoma.

**Table 4 cancers-15-00887-t004:** Univariate and multivariate analyses of local recurrence in tumor diameter ≤ 2.5 cm and >2.5 cm groups.

	Tumor Diameter ≤ 2.5 cm (*n* = 113)				Tumor Diameter > 2.5 cm (*n* = 91)			
	Univariate		Multivariate		Univariate		Multivariate	
	HR (95% CI)	*p*-Value	HR (95% CI)	*p*-Value	HR (95% CI)	*p*-Value	HR (95% CI)	*p*-Value
Age (per year)	1.0 (0.94–1.06)	0.88	0.97 (0.90–1.05)	0.48	0.97 (0.93–1.01)	0.18	0.96 (0.91–1.02)	0.20
Sex (male vs. female)	1.16 (0.36–3.75)	0.81	0.93 (0.28–3.09)	0.90	0.81 (0.34–1.94)	0.64	0.69 (0.29–1.67)	0.42
PS (2, 3 vs. 0, 1)	0.00004 (0.00002–0.0001)	<0.001	0.000031 (0.00001–0.00009)	<0.001	0.39 (0.053–2.87)	0.35	0.34 (0.044–2.57)	0.29
FEV_1_ (L) (≤1.5 vs. >1.5)	1.58 (0.52–4.80)	0.42	1.52 (0.49–4.72)	0.47	1.62 (0.69–3.81)	0.27	1.36 (0.58–3.20)	0.49
Tumor diameter (per 0.1 cm)	1.10 (0.99–1.23)	0.082	1.11 (0.98–1.25)	0.10	1.01 (0.95–1.07)	0.78	1.02 (0.95–1.09)	0.62
Histological type (SCC vs. ADC)	2.40 (0.79–7.3)	0.12	1.68 (0.59–4.76)	0.33	2.11 (0.89–5.04)	0.092	2.61 (1.07–6.41)	0.036
Biological effective dose (≤110 vs. >110)	1.52 (0.49–4.71)	0.47	1.54 (0.41–5.74)	0.52	1.01 (0.40–2.53)	0.99	0.80 (0.27–2.34)	0.69

PS = performance status, SCC = squamous cell carcinoma, ADC = adenocarcinoma.

## Data Availability

The data supporting this study are not publicly available due to the privacy of the research participants, but are available from the corresponding author on reasonable request.

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
