# Peer review of "Comparison of Recurrence Patterns between Adenocarcinoma and Squamous Cell Carcinoma after Stereotactic Body Radiotherapy for Early-Stage Lung Cancer"

_cancers, 2023, doi:10.3390/cancers15030887_

Round 1
Reviewer 1 Report
The Authors present a retrospective single-centre analysis of recurrence patterns of early NSCLC after SBRT considering histological differences. The paper is well written, with decent English. The methodology is clear and appropriate for the analyzed issue.
However, please find some remarks to be considered.
1. The novelty of the research is underlined; however, it is pretty well described in the literature. Nevertheless, it is still worth presenting your own “real-life” results.
2. Please specify how many recurrences were pathologically confirmed.
3. As it is challenging to distinguish the true recurrence from the late radiation-induced lung injury, please describe in detail recurrences diagnosed during the first year of the follow-up (the number, recurrence type, way of confirmation)
4. Could you please give more information about the morphology of local recurrences?
5. You found age and PS to be significant predictive factors. As they can be a surrogate of comorbidities, please specify why the patients were inoperable (name the comorbidities).
6. Please consider using the tumour location (peripheral/central) as another factor in the univariate and multivariable analyses.
7. Please elaborate on the discussion:
- It lacks the comparison of results from other studies.
- As DNA repair is mentioned as a potential explanation of failure, please comment on your fractionation schedule with a relatively long overall treatment time and refer to other studies.
- You describe the durvalumab consolidation as an option for future improvement – please briefly describe ongoing studies on that issue.
8. The references, in general, are adequate. However, they can be improved.
Reviewer 2 Report
Dear Authors,
thank you very mnuch for submitting your work to our journal. Here you describe the different pattern of recurrence between ADC and SCC in patients submitted to SBRT. the paper is substantially well written and clearly reported, however some issues need to be more deeply investigated.
- In the RT planning section you reported "[...]Planned doses of 44, 48, and 52 Gy 108 were prescribed for peripheral tumors with a maximum diameter of less than 1.5 cm, 1.5- 109 3 cm, and larger than 3 cm, respectively, until November 2008", so you delivered as much dose as larger was the target? Were you able to satisfy the constraints??Usually higher RT doses are delivered to smaller peripheral lesions. Please clarify.
- You mentioned that not all patients underwent 18FDG PET before being submitted to SBRT but PET18FDG is mandatory if patients are not submitted to broncoschopy or EBUS. So I think that speaking of pattern of recurrence it is mandatory to know how many pts underwent FDG PET, how many recurred and how many were ADC or SCC. These details may clarify the pattern of recucurrence partially solving this bias.
- In the backgroud you mentioned that SBRT is preferred ininoperable NSCLC patients but in the paper I did not find any information about this status. Because inoperable pts are usually frailer I think it's mandatory to know how many of these pts were judged inoperable or operable. It should also be important to report who defined a patient inperable (thoracic surgeon, MDT, etc etc). Inoperability may be a strong selection bias for this kind of study.
- In the reference you need to add a couple of recent paper that report significant information on yhis setting such as:
_ Current Radiotherapy tecnhiques in NSCLC:challenges and potential solutions. Giaj-Levra N, Borghetti P, Bruni A, Ciammella P, Cuccia F, Fozza A, Franceschini D, Scotti V, Vagge S, Alongi F. Expert Rev Anticancer Ther. 2020
_ Stereotactic ablative radiotherapy as an alternative to lobectomy in patients with medically operable stage I NSCLC: a retrospective multicenter analysis. Scotti V, Bruni A, Francolini G, Perna M, Vasilyeva P, Loi M, Simontacchi G, Viggiano D, Lanfranchi B, Gonfiotti A, Topulli J et al. Clin Lung Cancer, 2019
- At line 250 you affirm that dose escalation is recommended for stage III LA.NSCLC but I do not completely agree. Indeed the publication of phase III RCT RTOG 0617 showed a detrimental effect of dose escalation. Even if that trial had several bias, it has to be taken into aoccunt when speaking of this issue, So I think that you have to add some referral to that study and consequently add the linked reference in the bibliography. I think that actually dose escalation should have been investigated only in prospective clinical trial.
- In the conclusions you substantially report again the results you obtained. I think you have to revise the section summarizinh what previously reported and adding your considerations, for example, in terms of future perspectves.
Round 2
Reviewer 2 Report
Dear Authors,
thank you very much for having accepted our suggestions. The paper is certainly more complete and some issues were solved. However some more changes are needed.
In particular:
- line 80 - 81 : consist should be cahnged in "consisting"
- In the graph "Pretreatment evaluation" you need to report how many patients were staged by 18 FDG PET, how many by CT scan + bone scinitigraphy and how many by CT scan only (for this last subgorup you need to explain why, because it si not standard of care)
- all over the paper substitute "CT" with "CT scan"
- Line 189 : you reported that 21 of 77 recurrences were found within the first year but it does not represent the 10% as you wrote. Please clarify.
- Line 198 : what is the meaning of "comprehensive evaluation"? please clarify.
- English language needs to be reviewed by a native English, because even if fluent sometimes some grammar mistakes are present
